# Effect Evaluation and Action Mechanism Analysis of “Profile Control + Plugging Removal” after Chemical Flooding

**DOI:** 10.3390/gels8070396

**Published:** 2022-06-22

**Authors:** Jianchong Gao, Xiangguo Lu, Xin He, Jinxiang Liu, Kaiqi Zheng, Weijia Cao, Tianyu Cui, Huiru Sun

**Affiliations:** 1Key Laboratory of Enhanced Oil and Gas Recovery of Ministry of Education, Northeast Petroleum University, Daqing 163318, China; hi8457918@yeah.com (J.G.); hexin_831@163.com (X.H.); liujx_118@163.com (J.L.); 13354597101@sina.cn (K.Z.); caoweijia131466@163.com (W.C.); 1810312443@qq.com (T.C.); 3273804832@qq.com (H.S.); 2Liaodong Operation Area of Tianjin Branch of CNOOC, Tianjin 300452, China

**Keywords:** JZ9-3 oilfield, general plugging removal, profile control + plugging removal, EOR, mechanism analysis

## Abstract

The existing plugging removal operation in JZ9-3 oilfield has the disadvantages of small amount of plugging remover, fast injection speed, and short construction time. Under the condition of injection well suction profile reversal, plugging remover is difficult to enter the low permeability part and play the role of deep plugging removal. In order to improve the plugging removal effect, this paper used the physical simulation method to carry out the experimental study and mechanism analysis on the effect of water flooding, chemical flooding, and plugging removal measures of the multi-layer system combination model. The results showed that the recovery of general plugging removal after chemical flooding increases by only 0.70%, while the recovery of ‘profile control + plugging removal’ increases by ‘9.34% + 2.59%’, and the amount of produced liquid decreases by more than 40%. It can be seen that the combined operation of profile control and plugging removal has dual effects of plugging and dredging and synergistic effect, which not only expands the swept volume, but also reduces the inefficient and ineffective cycles. On this basis, the optimization design and effect prediction of the target well W4-2 plugging removal scheme were carried out by using the numerical simulation method. Recommended scheme: inorganic gel profile control agent volume 13,243.6 m^3^, produced by the main agent (Na_2_O·nSiO_2_), isolation fluid (Water), and auxiliary agent (CaCl_2_) through multiple rounds of alternating injection into the reservoir. The plug removal agent (K_2_S_2_O_8_) injection volume is 100 m^3^, the concentration is 0.8%. The post-implementation ‘Output/Input’ ratio is expected to be 3.7.

## 1. Introduction

Although chemical flooding in the JZ9-3 oilfield has achieved a good effect on oil increase and precipitation, the water wells in the middle and later stages of chemical flooding and subsequent water flooding have experienced “Reverse rotation of liquid absorption profile”, resulting in decreased liquid absorption and intensified inefficient and ineffective circulation [1]. “Liquid absorption profile reversal” means that the middle and low permeability layers (parts) of the reservoir are seriously damaged due to polymer retention. In order to remove the blockage and improve the liquid absorption profile, the oil field has successively used acid, hydrogen peroxide, biological enzyme, solid chlorine dioxide, benzoyl peroxide, ammonium persulfate, and potassium persulfate to remove the blockage of water wells. Good injection increase effect has been seen in the initial stage, but the maintenance time is short, and the oil increase effect at the oil well end is generally poor [2,3,4,5,6,7,8,9,10]. As the liquid suction pressure difference, i.e., the liquid suction capacity, at the high permeability part of the reservoir is much greater than that at the middle and low permeability part, and in addition, the plugging removal dose used in the construction operation is less, the liquid injection speed is faster, and the construction time is shorter, the plugging removal agent suction at the middle and low permeability part of the reservoir is very small, and the plugging removal effect is poor [11,12,13,14,15,16]. In the oilfield production practice, increasing the liquid suction pressure difference is the only way to increase the suction amount of plugging removal agents in the middle and low permeability parts and improve the plugging removal effect. Increasing the injection pressure is an important measure to increase the liquid suction pressure difference [17,18,19,20,21,22]. Increasing the injection pressure can be achieved by increasing the injection speed and increasing the seepage resistance at the high permeability part, i.e., the start-up pressure of liquid absorption. The former will have a poor effect due to the influence of process conditions and the “flow around” of fluid in the reservoir, while the latter can increase the injection pressure without increasing the injection speed [23]. Therefore, “profile control + plugging removal” is one of the effective technical ways to further enhance oil recovery after chemical flooding. In order to realize the effective plugging removal of the middle and low permeability layer in the polymer plugging well, according to the actual situation existing in the production process of the JZ9-3 oilfield, the parallel core physical simulation experiment was used to explore the influence of the flooding agent injection mode, the injection pressure of the depolymerization system and the joint operation of “profile control + plugging removal” on the plugging removal effect. The scheme design and effect prediction are carried out for difficult wells. The slug size of the profile control agent is optimized and the economic benefit is evaluated by using numerical simulation technology. Good results are obtained in the experimental process, which improves the technical support for the application of plugging removal process measures in oilfields.

## 2. Results and Discussion

### 2.1. Polymer Rheology

A salt-resistant polymer with a molecular weight of 19 million and a concentration of 2000 mg/L was prepared with simulated injection water. The rheological properties of the polymer under reservoir temperature were tested. See Figure 1 for the rheological properties of the polymer.

It can be seen from Figure 1 that the apparent viscosity of the polymer decreases with the increase of the shear rate. The polymer has good rheological properties at reservoir temperature.

### 2.2. Polymer FTIR

The prepared polymer powder was mixed with spectroscopic pure KBr to make tablets, which were determined by spectrum one infrared spectrometer. See Figure 2 for the infrared spectrum of polymer.

It can be seen from Figure 2 that the wave number 3340 cm^−1^ is the characteristic absorption peak of the primary amide-NH_2_ group, the wave number 2937 cm^−1^ is the stretching vibration absorption peak of the secondary amide NH group, the wave number 1680 cm^−1^ is the stretching vibration absorption peak of −C=O in the amide group, and the wave number 1580 cm^−1^ is the characteristic stretching vibration peak of the carboxyl group, indicating that the hydrolysis reaction has taken place, and the wave number 1055 cm^−1^ is the asymmetric absorption peak of −SO_3_ of the sulfonic acid group. From the analysis of the infrared spectrum, it can be seen that there are characteristic absorption peaks of the carboxyl group, amide group, and sulfonic group in the polymer, and there is no −C=C− double bond in the molecular structure. It shows that some amide groups of the polymer have been hydrolyzed, and there are salt-resistant monomers, which is a salt-resistant polymer.

### 2.3. Evaluation of Basic Properties of Inorganic Gel

Sodium silicate and Calcium chloride are prepared with simulated injection water from the J oilfield. A total of 50 mL of each is added to a 100 mL measuring cylinder and placed in a 65 °C incubator. The relationship between the standing time and the volume of inorganic gel is shown in Figure 3.
Na_2_SiO_3_ + CaCl_2_ = CaSiO_3_ ↓ + 2NaCl(1)

It can be seen from Figure 3 that the main agent reacts with the auxiliary agent to form an inorganic gel. With the increase of the standing time, the volume of inorganic gel gradually decreases. The analysis shows that the inorganic gel reaction “coagulates at one touch”. With the increase of the standing time, the inorganic gel settles, which can form an inorganic coating in the porous medium, reduce the overflow section, and promote the subsequent plugging removal system to enter the medium and low permeability layer. At the same time, sodium silicate is alkaline, which can convert the amino group of the polymer into a carboxyl group, and further increase the flow capacity of the polymer.

### 2.4. Influence of Injection Method of Oil Displacement Agent on Fractional Flow Rate, Recovery Factor, and Plugging Removal

(1)Recovery

Experimental data on influence of polymer solution injection mode on chemical flooding and plugging removal effect are shown in Table 1.

Can be seen from Table 1, in “constant speed” (1-1) under the experimental conditions, with the increase of polymer solution, injected slug size, the higher injection pressure, low permeability model parts (including “high permeability core” and “low permeability of low permeability cores” part, hereinafter the same) absorb hydraulic and fluid volume increases, the swept volume increase, increase recovery factor. Compared with the “constant speed” injection method, the “constant speed and constant pressure” injection method is limited by injection pressure. In the process of chemical flooding, the suction pressure difference and the increase of suction volume in low permeability parts of the model decrease, and the effect of expanding sweep volume becomes worse, so the increase of recovery rate decreases from 23.18% to 20.12%. Compared with “constant speed” injection, “constant speed and constant pressure” injection improves the plugging effect, but not much. Further analysis shows that compared with the “high permeability core”, the “low permeability core” is the main potential area for further enhanced oil recovery after chemical flooding due to its small increase in the recovery rate and high remaining oil saturation.

(2)Dynamic characteristics

During the experiment, the relationship between injection pressure, water cut, and recovery, and PV is shown in Figure 4.

As can be seen from Figure 4, compared with the “constant speed and constant pressure” injection mode, the “constant speed” injection pressure in the chemical flooding stage is higher, and the suction hydraulic pressure difference and liquid suction volume increase in low permeability parts of the model are larger, and the effect of expanding sweep volume is better. It can be seen from 2 that the greater the increase of injection pressure, the greater the increase of chemical flooding recovery. In the field chemical flooding practice, the injection pressure is limited by the fracturing pressure of reservoir rock, and the changing trend of injection pressure of the “constant velocity and constant pressure” injection method is similar to the actual reservoir, so the experimental recovery value obtained by using this injection method can more objectively reflect the actual oil increase and precipitation effect of the reservoir.

(3)Diversion rate

The relationship between the diversion rate and PV at the produced end of the model in each displacement stage during the fractional flow rate experiment is shown in Figure 5.

As can be seen from Figure 5, compared with the “constant velocity and constant pressure” injection mode, the “constant velocity” injection mode has a larger value of the “high permeability core” and a smaller value of the “low permeability core” in the chemical flooding and subsequent water flooding stages. It can be seen that higher injection pressure in the chemical flooding stage is not conducive to the expansion of sweep volume of the “low permeability core”.

### 2.5. Influence of Plugging Agent Injection Pressure on Plugging Effect

(1)Recovery

Experimental data on the influence of plugging agent injection pressure on the plugging effect after chemical flooding are shown in Table 2.

As can be seen from Table 2, with the increase in plugging agent injection pressure (2.5P, 3.0P and 3.5P), the plugging effect increased (oil recovery increased by 0.69%, 0.74% and 0.88%). Analysis, by increasing injection pressure, can increase model in low permeability of three hydraulic differential absorption liquid and absorption, but the high permeability area hydraulic differential absorption and absorption and additional fluid volume and the growth of larger, combined with the solution of plugging agent to low permeability in the reservoir area flow around to high permeability area, eventually reduce the effect of deep low permeability parts broken down. It can be seen that increasing injection pressure has no obvious effect on improving the plugging effect.

(2)Dynamic characteristics

In the experimental process, the relationship between injection pressure, liquid production rate, and recovery, and PV is shown in Figure 6.

As can be seen from Figure 6, with the increase in injection pressure, the plugging dose of the “high permeability core” and high permeability part of the model is much higher than that of the “low permeability core” and low permeability part. The plugging effect of the former is significantly better than that of the latter, and the extraction speed of the former is also higher than that of the latter. It can be seen that increasing the injection pressure of the plugging agent will increase the reservoir heterogeneity, increase the fluid absorption amount of “high permeability core” and high permeability parts, and aggravate the low efficiency and invalid circulation, which is not conducive to expanding the sweep volume effect.

(3)Diversion rate

The relationship between the diversion rate and PV at the produced end of the model at each stage of the experiment is shown in Table 3.

As can be seen from Table 3, with the increase of plugging agent injection pressure, the divergence rate of the produced end of the model “high permeability core” increases, while that of the model “low permeability core” decreases, but the variation range is not large. Analysis thinks, that although increased injection pressure increases “high permeability core” and “low permeability core” in the middle and lower permeability site blocking agent hydraulic differential settlement to inhale uptake, plugging agent solution will happen in the heterogeneous cores “cross infiltration” function that is from the low permeability parts flow around to high permeability area, thus model production end diversion rate changed little. Therefore, it is an effective way to reduce the “cross-seepage” effect of each position, increase the action range of the plugging agent and improve the plugging effect by increasing the seepage resistance of the high permeability position of the reservoir.

### 2.6. Effect of Combined Operation of “Profile Control + Plugging Removal”

(1)Recovery

Experimental data on oil-increasing and precipitation effect of the chemical flooding model “profile control + plugging removal” are shown in Table 4.

As can be seen from Table 4, the recovery rate increased by 20.12% by chemical flooding alone and 0.69% by plugging alone (Plan 3-1).Chemical flooding and “blockage” profile control + recovery efficiency increases 27.82% and 32.31%, among them “blockage” profile control + recovery growth “6.18% + 1.41%” (3-2) and “9.34% + 2.59%” (3-3).It can be seen that the oil increase effect of the combined operation of profile control and plugging removal is significantly higher than that of simple plugging removal. The larger the slug size of the profile control agent is, the better the effect of “profile control + plugging removal” on oil increase and precipitation is.

(2)Dynamic characteristics

In the experimental process, the relationship between injection pressure, production speed, and recovery, and PV is shown in Figure 7.

As can be seen from Figure 7, due to the good injection performance of the inorganic gel profile control agent and the auxiliary measures to reduce injection pressure by reducing injection speed, the profile control agent only enters the high permeability parts of model “high permeability core” and “low permeability core” and generates inorganic gel, thus increasing seepage resistance and suction starting pressure. In the follow-up plugging process, not only does the amount of plugging agent absorbed at the injection end of each core in the model increase, but also the “cross-permeability” effect between various permeable parts of the core is obviously weakened, the action range of plugging agent is enlarged, and the plugging effect is improved, so as to expand the sweep volume and improve the recovery rate. Further analysis shows that compared with simple plugging removal (3-1), “profile control + plugging removal” (3-2 and 3-3) can effectively inhibit low efficiency and invalid circulation of injected water after chemical flooding 5, reduce the liquid production speed by more than 40%, and significantly reduce the treatment cost of produced liquid.

(3)Diversion rate

The relationship between the diversion rate and PV at the produced end of the model at each stage of the experiment is shown in Figure 8.

As can be seen from Figure 8, compared with general plugging after chemical flooding, the divergence rate of “high-permeability core” decreases, and that of “low-permeability core” increases after adopting “profile control + plugging”. The larger the slug size of the profile control agent is, the higher the divergence rate of the “low permeability core” is, and the better the effect of expanding sweep volume is.

### 2.7. Screening of Target Wells

In recent 3 years, the plugging operation parameters and effect statistics of some chemical flooding and water flooding injection Wells are shown in Table 5.

It can be seen from Table 5 that 15 Wells of chemical flooding and water flooding injection Wells in the JZ9-3 oil field were successfully de-plugging from 2018 to 2020, with a short term validity and poor de-plugging effect. Among them, Wells W4-2 were de-plugging twice on 15 September 2019 and 19 March 2020 respectively, both of which failed to achieve the expected de-plugging effect. It belongs to the difficult well of plugging operation, so it is selected as the candidate well for “profile control + plugging” construction scheme design.

### 2.8. Injection Parameter Optimization and Process Scheme Design of Well W4-2

#### 2.8.1. Geological Modeling and Rock Fluid Parameters

CMG software was used for numerical simulation. The geological model is divided into rectangular grids. The grid number of the local numerical model of well W-4-2 is 83 × 43 × 17 = 61,564, and the plane grid size is 10 m × 10 m. Parameters such as thickness, porosity, permeability, and oil saturation of each layer were obtained from field test data, and the production history of the well since it was put into production was fitted. The rock and fluid property parameters of the local polymer injection block in well W-4-2 are set in Table 6. The relative permeability curve is obtained from the data provided by the oil field, which determines the law of oil-water flow in the process of numerical simulation. Oil-water relative permeability curve is shown in Figure 9.

It can be seen from Figure 9 that with the increase in water saturation, the oil phase permeability decreases, and the water phase permeability increases. The rock surface is weakly lipophilic, with large permeability and small capillary force.

#### 2.8.2. Influence of Depolymerizer Injection Mode on Plugging Removal Effect

On 3 March 2020, 80 m^3^ depolymerization agent was injected into target well W4-2 for 6 h of injection time and 10 h of soaking. The average polymer concentration in the area near-wellbore before and after plugging in well W4-2 is shown in Table 7, and the relationship between injection speed and time is shown in Figure 10.

It can be seen from Table 7 and Figure 10 that compared with “general” plugging, the average polymer concentration of the high-permeability layer decreases less after plugging by “profile control + plugging”, while that of the low-permeability layer decreases more. Similarly, after “profile control + plugging removal”, the average permeability of the whole well decreases, the liquid absorption rate decreases, the permeability level difference in each part of the reservoir decreases, and the effect of expanding the sweep volume becomes better. It can be seen that the combined operation of “profile control + plugging” can not only improve the liquid absorption amount of the low permeability layer (part) and expand the sweep volume but also reduce the liquid absorption amount of the high permeability layer (part) and reduce the phenomenon of low efficiency and invalid circulation, with obvious technical and economic benefits.

#### 2.8.3. Influence of Depolymerizer Injection Amount on Plugging Removal Effect

Under the condition of 0.6%, see Table 8 for the experimental results of the influence of plugging removal agent injection amount on the plugging removal effect.

As can be seen from Table 8, the amount of oil added increases with the increase of the injection amount of the depolymerization system. When the injection rate exceeds 100 m^3^, the increase rate decreases. It can be seen that the reasonable amount of depolymerization system in well W4-2 is about 100 m^3^.

#### 2.8.4. Influence of Injection Concentration of Depolymerizer on Plugging Removal Effect

When the amount of depolymerization agent is 100 m^3^, the experimental results of the influence of the concentration of depolymerization agent on the plugging effect are shown in Table 9.

It can be seen from Table 9 that with the increase of the concentration of depolymerization agent, the amount of oil increased and the plugging effect improved. When the concentration of depolymerization agent exceeds 0.8%, the increase rate of plugging increase decreases obviously. Therefore, the reasonable concentration of the depolymerization agent should exceed 0.8%.

#### 2.8.5. Influence of Soaking Time on Plugging Removal Effect

Under the condition of 300 m^3^ depolymerization agent dosage and 0.8%, the influence of soaking time on the plugging effect is shown in Table 10.

It can be seen from Table 10 that with the increase in soaking time, the oil increase first increases and then flattens out after 12 h. There is little difference between the oil increase for 48 h and 12 h after soaking. Therefore, the recommended soaking time is 12 h~24 h.

### 2.9. Economic Benefit Prediction of “Profile Control + Plug Removal” Technology

When calculating the “output/input” ratio, the crude oil futures price is 60 USD/barrel, and the exchange rate is 1 USD = 6.36 RMB. A total of 1 m^3^ crude oil equals 6.2893 barrels, and the crude oil price is 6.2893 × 60 × 6.36 yuan/m^3^ = 2400 yuan/m^3^. Profile control agent includes main agent A and auxiliary agent B. The injection concentration of the profile control agent is 4%, and the injection volume is calculated according to 4500 m^3^. The depolymerization system mainly includes a depolymerization agent, corrosion inhibitor, and dispersant. See Table 11 for details of pharmaceutical and construction prices.

According to the previous field practice experience, profile control radius is set as 1/3 well spacing, i.e., 125 m, plane sweep coefficient is 0.5, and water flooded thickness coefficient is 0.5. W4-2 well I oil combination II oil group high permeability layer thickness 2.5 m and 3.7 m, porosity 28.6%, and 27.8%, remaining oil saturation 35.5% and 35.5%, calculated profile control agent dosage 13,243.6 m^3^, through calculation of the “output/input” ratio of each scheme. The calculation results show that under the conditions of 13,243.6 m^3^ of profile control agent, 100 m^3^ of plugging agent, and 0.8%, the “output/input” ratio is expected to be 3.7 after the implementation of profile control + plugging solution, which is the highest value in all schemes. Volume method was adopted to calculate the dosage of the profile control agent:(2)$$V=\alpha \pi R^{2}fh\varphi \left(1-S_{o}\right)$$*V*-Profile control agent dosage, m^3^;A-Plane conformance (A = πR^2^);*R*-Radius of profile control, m;*f*-Flood thickness coefficient;*h*-The thickness of the profile, m;*φ*-porosity, %;*S**_o_*-Oil saturation.


### 2.10. Mechanism Analysis of “General Plugging Removal” and “Profile Control + Plugging Removal”

The ultimate goal of “general plugging removal” and “profile control + plugging removal” is to enhance oil recovery. The action mechanism of “general plugging removal” and “profile control + plugging removal” is shown in Figure 11 and Figure 12.

It can be seen from Figure 11 and Figure 12 that in the process of polymer flooding, firstly, the polymer enters the high-permeability layer with low seepage resistance and stays in the pore throat (chemical adsorption and mechanical capture), resulting in the increase of seepage resistance in the high-permeability layer, the decrease of suction pressure difference and the increase of injection pressure. When the injection pressure is greater than the start-up pressure of liquid absorption in the medium and low permeability layer, the medium and low permeability layer begins to absorb liquid, that is, the fluid flow turns in the reservoir. At the same time, the seepage resistance of the middle and low permeability layer increases, but the increase rate of the seepage resistance is greater than that of the high permeability layer. At this time, the suction profile is reversed, and the high permeability layer starts to suck again. It is worth noting that the polymer will cause damage to the medium and low permeability layer after entering the medium and low permeability layer. Therefore, it is necessary to carry out plugging removal for the medium and low permeability layer. The commonly used plugging removal method in the oil field is “general plugging removal”. That is, in the late stage of polymer flooding, plugging remover is directly injected, and the plugging remover enters the high permeability layer along the part with relatively small seepage resistance. During the well plugging process, it reacts with the polymer retained in the high permeability layer, further reducing the seepage resistance of the high permeability layer and increasing the subsequent inefficient and ineffective circulation of water in the high permeability layer. “Profile control + plugging removal” means that after polymer flooding, an inorganic gel with a small slug size is alternately injected as a profile control agent to form a small range of plugging near the well zone of the high permeability layer, which increases the seepage resistance at the inlet of the high permeability layer and near the well zone, the injection pressure, the liquid absorption pressure difference in the middle and low permeability layer, and the relative liquid absorption volume. Then inject the plugging removal agent to make it oxidize and degrade with the polymer retained in the plugging part near the well in the middle and low permeability layer. The long chain of the polymer is oxidized and degraded into a short chain, which increases the liquid flow capacity of the middle and low permeability layer, further reduces the seepage resistance of subsequent water in the middle and low permeability layer, and increases the liquid suction pressure difference and relative liquid suction capacity of the middle and low permeability layer.

## 3. Conclusions

Increasing the injection pressure can increase the liquid suction pressure difference in the low and middle permeability layer of the reservoir and the suction amount of the plugging removal agent, but the liquid suction pressure difference and the suction amount in the high permeability layer also increase correspondingly and increase greatly, so increasing the injection pressure does not significantly improve the plugging removal effect.The current general plugging removal operation uses less plugging removal agent, and the liquid injection speed is faster, resulting in less suction of plugging removal agent at the middle and low permeability parts. After plugging removal, the permeability differential is further increased, which intensifies the low efficiency and invalid circulation.Compared with the simple general plugging removal operation, the combined operation of “profile control + plugging removal” has the dual effects of “plugging” and “drainage”. The experiment shows that the increase of general plugging removal recovery is only 0.70%, the increase of “profile control + plugging removal” recovery is “9.34% + 2.59%”, and the produced fluid volume is reduced by more than 40%. Good injection and plugging performance of profile control agent is the technical guarantee for the success of “profile control + plugging removal” joint operation.Well W4-2 is recommended as the field test target well for “profile control + plug removal”. The test scheme includes 13,243.6 m^3^ profile control agent, 100 m^3^ plug removal agent and 0.8%. It is estimated that the “output/input” ratio will be 3.7 after the implementation of the scheme.

## 4. Material and Methods

### 4.1. Experimental Materials

In order to simulate the plugging problem of salt-resistant polymer in the offshore oilfield, the salt-resistant polymer used in the JZ9-3 oilfield is used in the experiment. The polymer is polymer SNF3640 with effective content of 90% and a molecular weight of 19 million, which is provided by the CNOOC Tianjin Branch, Tianjin, China. The profile control agent is composed of “4% auxiliary agent (calcium chloride) solution, 4% main agent (sodium silicate) solution and isolation fluid (water)” injected alternately to produce inorganic gel [18,19,20] in the core, and the isolation fluid is JZ9-3 oilfield simulated injection water. The plug removal agent is composed of “0.1% depolymerizing agent +0.5% corrosion inhibitor + 1.0% dispersant”.

The oil used for the experiments was a simulated oil, made from JZ9-3 field crude oil and white oil in proportion to each other, with a viscosity of 18 mPa·s at a reservoir temperature of 60 °C. The water used for the experiments was simulated injection water from the JZ9-3 oilfield, and the water quality analysis is shown in Table 12.

The experimental model of “profile control + plugging removal “consists of a “high permeability core” and a “low permeability core” in parallel, which are made [24,25,26] of quartz sand epoxy resin cemented and compressed. Considering that the JZ9-3 oilfield adopts a multi-layer system (oil group) development model with a well network, among which oil groups I and II are the main oil groups and are also the main sorbent layers of chemical drive, the unblocking also mainly involves this oil group. The permeability parameter design of this experimental model mainly refers to the logging interpretation results of multiple injection wells. Among them, the “high permeability core” simulates the “I oil group”, which includes three equal thickness layers of “high/medium/low”. *K*_g_ = 4500/2500/500 × 10^−3^ μm^2^; “low permeability core” simulates “II oil group”, each sublayer *K*_g_ = 1000/500/250 × 10^−3^ μm^2^. Core geometric size: width × height × length = 4.5 cm × 4.5 cm × 30 cm.

### 4.2. Apparatus and Experimental Procedures

(1)Instrumentation

The experimental apparatus for evaluating the effectiveness of chemical drive and “profile control + plugging removal “includes components such as advection pumps, pressure sensors, intermediate vessels, and pipeline gates, all of which are placed in a 60 °C incubator except for the advection pumps and hand pumps. The flow of the experimental equipment is shown in Figure 13.

(2)Experimental steps
The saturated water is extracted from the core to measure the pore volume and porosity;“High permeability core” and “low permeability core” are saturated with oil respectively to calculate oil saturation;The “high permeability core” and “low permeability core” are connected in parallel to form a model. The model is water-driven until the water cut is 80%, and the injection pressure P at the end of water flooding is recorded;Inject polymer solution into the model until the slug size reaches 0.5 PV;Subsequent water flooding of the model to a water cut of 90%;Transferring inorganic gel profile control agent;Inject plugging removal agent into the model, and block the well for 12 h;The subsequent water flooding of the model reaches a water cut of 98%. During these experiments, the model injection pressure and the volume of fluid injected and extracted from each sub-layer were regularly recorded, after which the recovery rate, water cut, and sub-layer diversion rate were calculated and the injection pressure, water cut, recovery rate, sub-layer diversion rate and recovery rate versus PV were plotted. The experimental design is shown in Table 1. Where “P” is pressure after water flooding.

### 4.3. Scheme Design

To simulate the plugging of polymer injection in the early stage of the Oilfield, JZ9-3 field salt resistant polymer (1900 × 10^4^, 1200 mg/L, 15 mPa·s) is injected after water flooding to 80% water cut, and then the subsequent water flooding to 90%. After injection of plugging removal agent, the well is blocked for 12 h, and then the subsequent water flooding is carried out until the water cut is 98%. See Table 13 for the experimental scheme design. The reaction speed of the inorganic gel main agent and the auxiliary agent is fast. In order to avoid the blockage of inorganic gel on the end face of core, the method of adding an isolation liquid slug is adopted to inject inorganic gel alternately. Inject 0.06 PV of 4% main agent, 0.01 PV of isolation solution (calcium and magnesium ions removed), 0.06 PV of 4% auxiliary agent, and so on for 5 rounds.

## Figures and Tables

**Figure 1 gels-08-00396-f001:**
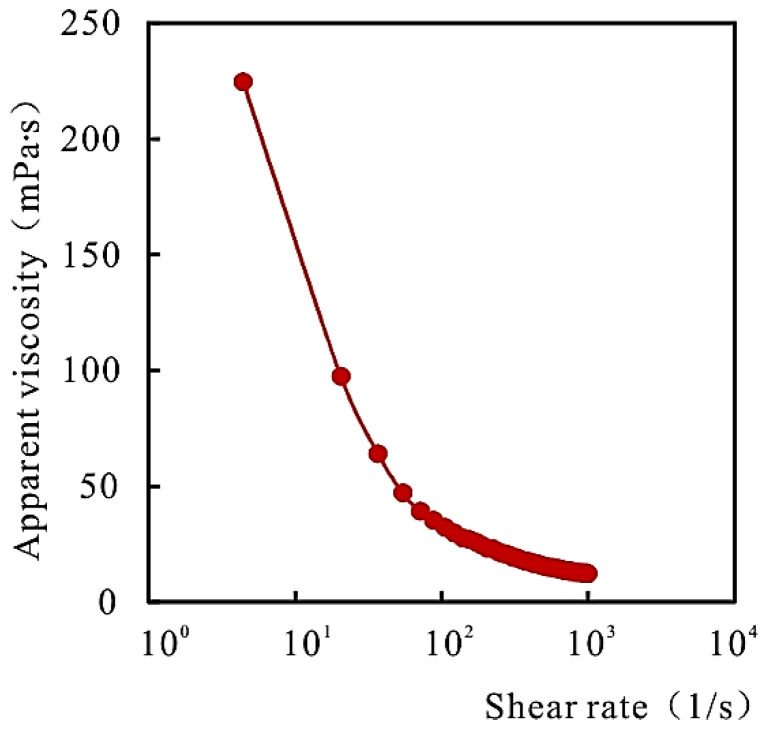
Relationship between polymer shear time and apparent viscosity.

**Figure 2 gels-08-00396-f002:**
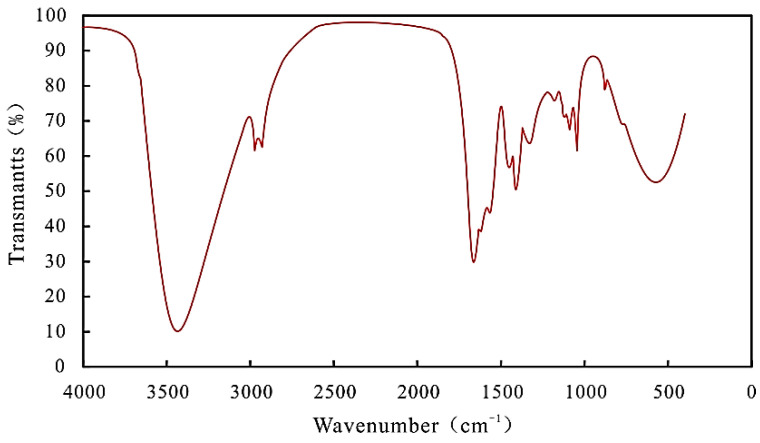
Infrared spectrum of polymer.

**Figure 3 gels-08-00396-f003:**
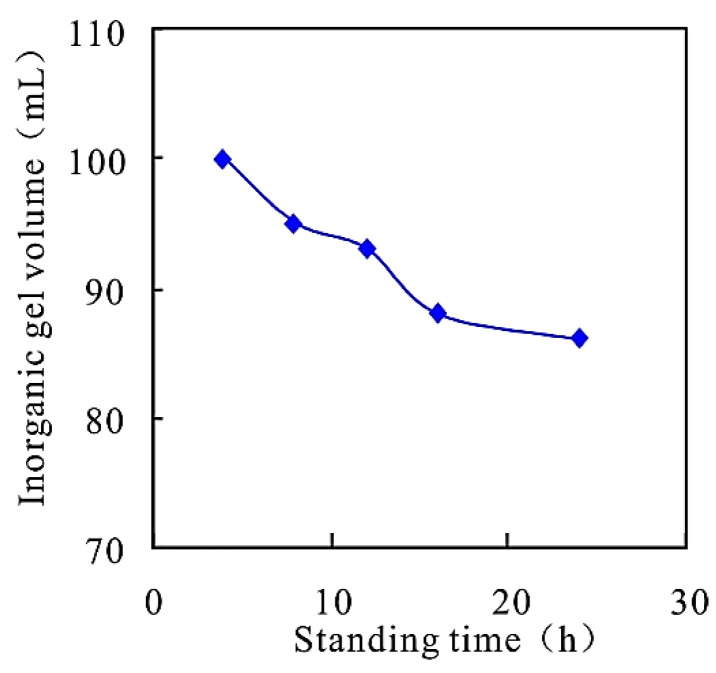
Relationship between standing time and inorganic gel system.

**Figure 4 gels-08-00396-f004:**
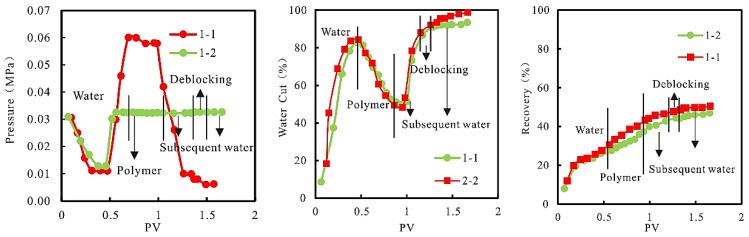
Relationship between injection pressure, water cut, and recovery and PV.

**Figure 5 gels-08-00396-f005:**
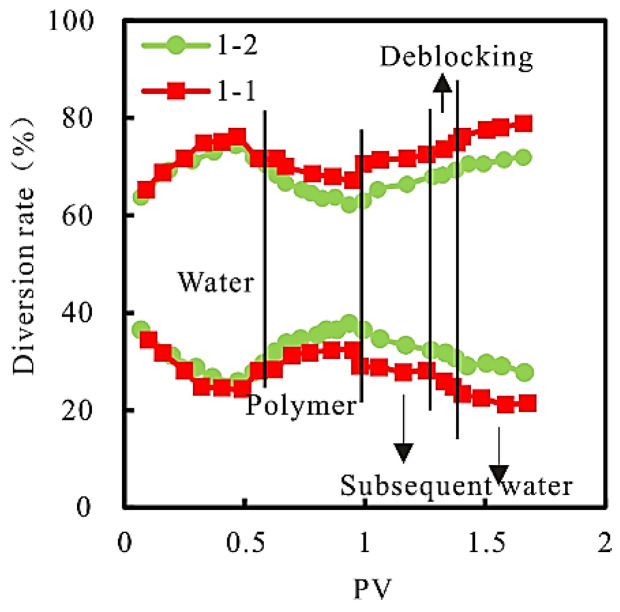
Relationship between diversion rate and PV at production end.

**Figure 6 gels-08-00396-f006:**
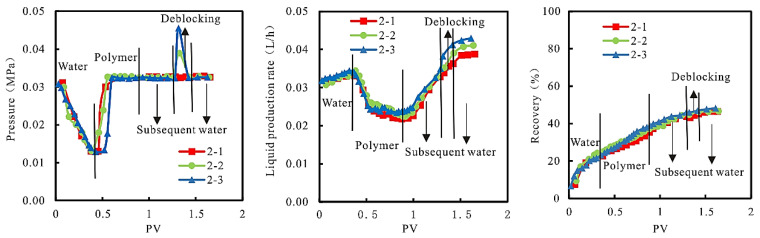
Relationships between injection pressure, liquid production rate, and recovery, and PV.

**Figure 7 gels-08-00396-f007:**
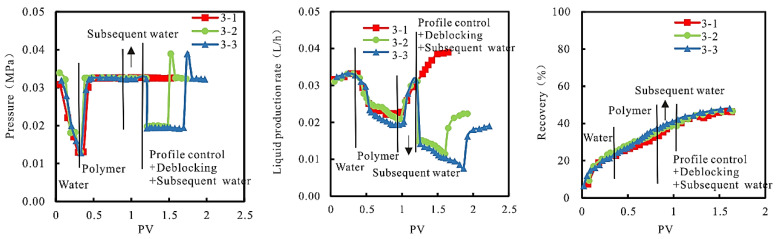
Relationships between injection pressure, production rate, and recovery, and PV.

**Figure 8 gels-08-00396-f008:**
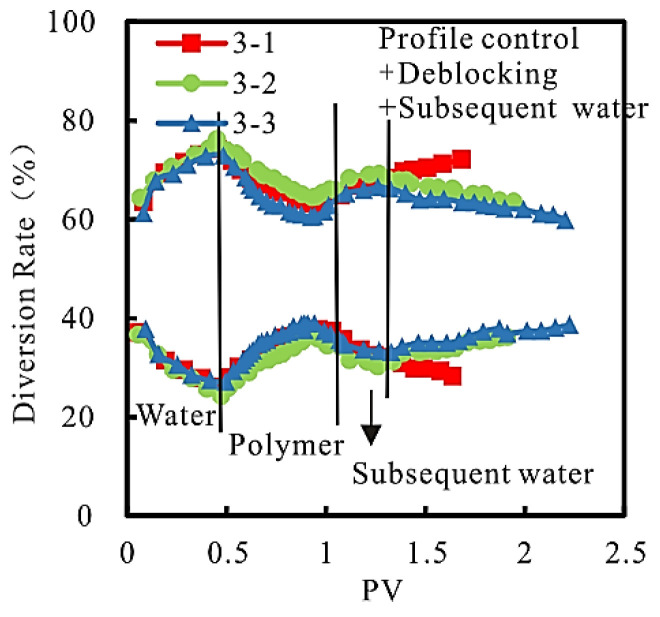
Relationship between diversion rate and PV.

**Figure 9 gels-08-00396-f009:**
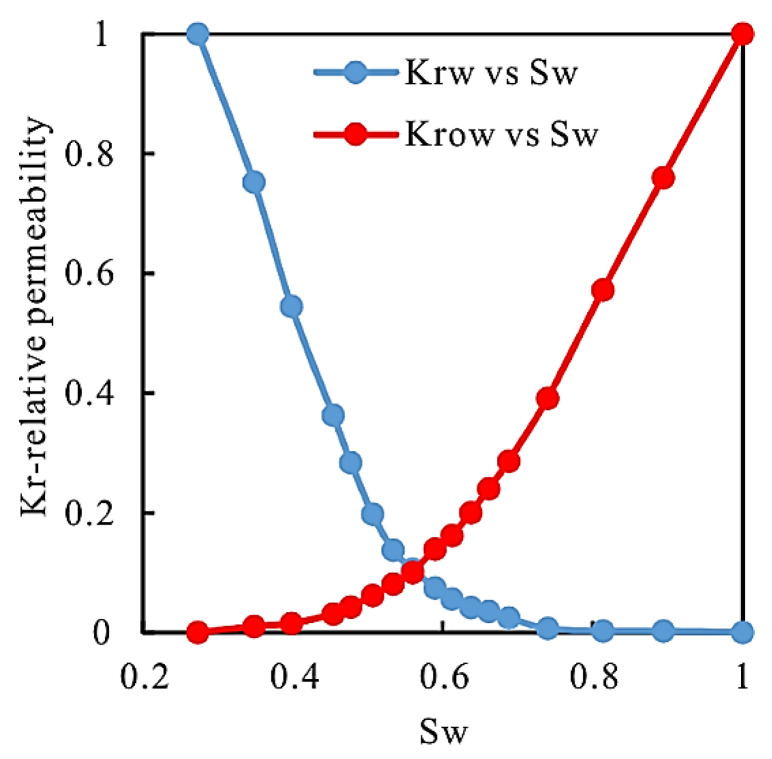
Oil-water relative permeability curve.

**Figure 10 gels-08-00396-f010:**
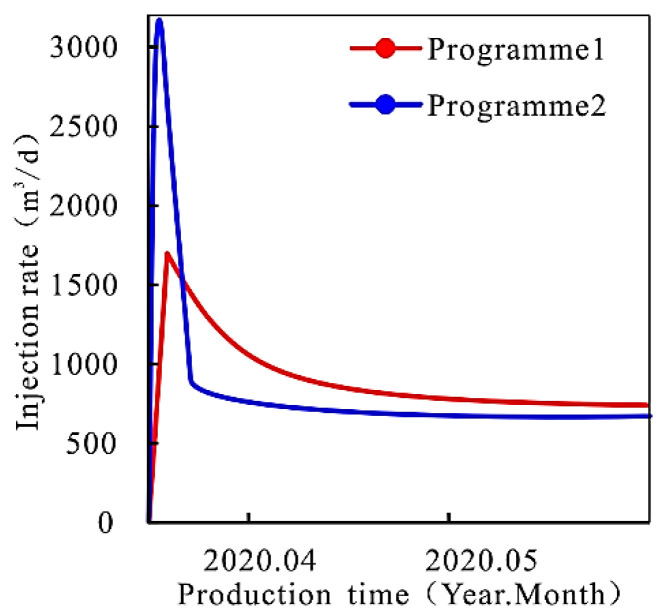
Relationship between injection speed and time.

**Figure 11 gels-08-00396-f011:**
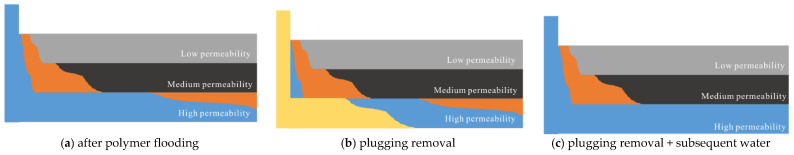
General plugging removal mechanism (**a**) after polymer flooding, (**b**) plugging removal and (**c**) plugging removal + subsequent water flooding.

**Figure 12 gels-08-00396-f012:**
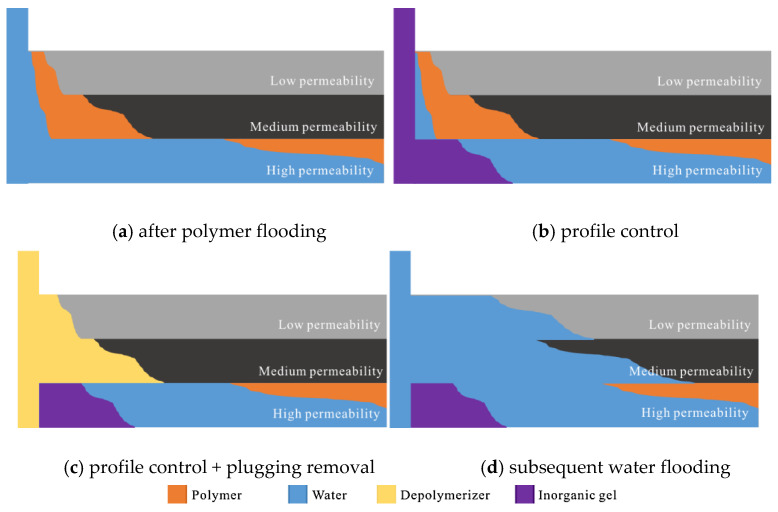
Action mechanism of “profile control + plugging removal” (**a**) after polymer flooding, (**b**) profile control, (**c**) profile control + plug removal and (**d**) subsequent water flooding.

**Figure 13 gels-08-00396-f013:**
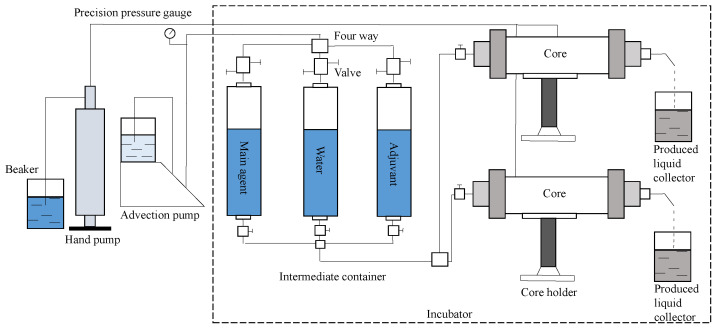
Schematic diagram of the experimental equipment and process.

**Table 1 gels-08-00396-t001:** Experimental data of recovery factor.

	Paramete	Oil Saturation (%)	Recovery (%)			Recovery Factor Increase (%)	
Package Number			Water Flooding	Polymer Flooding Stage	Broken Down Stage	Broken Down Stage	The Overall
1-1	High permeability cores	70.89	36.00	64.67	65.27	0.60	29.27
	Low permeability core	70.01	8.55	18.71	19.29	0.58	10.74
	model	70.57	26.80	49.39	49.98	0.59	23.18
1-2	High permeability	71.51	36.10	62.56	63.26	0.70	27.16
	Low permeability	70.75	8.34	15.24	15.92	0.68	7.58
	model	70.49	26.22	45.65	46.34	0.69	20.12

**Table 2 gels-08-00396-t002:** Experimental data of recovery.

	Parameter	Oil Saturation (%)	Recovery (%)			Recovery Factor Increase (%)	
Package Number			Water Flooding	The Chemical Flooding	Broken Down Stage	Broken Down Stage	The Overall
2-1 (2.5P)	High permeability cores	71.51	36.10	62.56	63.26	0.70	27.16
	Low permeability core	70.75	8.34	15.24	15.92	0.68	7.58
	model	70.49	26.22	45.65	46.34	0.69	20.12
2-2 (3.0P)	High permeability cores	71.87	36.46	63.95	64.70	0.75	29.24
	Low permeability core	70.62	8.23	15.10	15.82	0.72	7.59
	model	71.43	26.09	46.08	46.82	0.74	20.73
2-3 (3.5P)	High permeability cores	71.30	37.20	64.48	65.42	0.94	28.22
	Low permeability core	70.85	7.86	16.20	16.52	0.78	8.66
	model	70.96	26.57	46.93	47.81	0.88	21.24

**Table 3 gels-08-00396-t003:** Experimental data of diversion rate in each stage of small layer.

	Parameter	Model of Core	Permeability *K*_w_ (10^−3^ μm^2^)	Stage Fractional Flow Rate (%)				
Package Number				Water Flooding End	End of the Polymer Injection	Subsequent Water Flooding Is Complete	Note Plugging Agent End	Subsequent Water Flooding Is Complete
2-1 (2.5P)		High permeability cores	1231	74.2	62.4	69.1	70.2	71.8
		Low permeability core	349	25.8	37.6	30.9	29.8	28.2
2-2 (3.0P)		High permeability cores	1235	74.3	62.3	69.7	70.8	72.5
		Low permeability core	367	25.7	37.7	30.3	29.2	27.5
2-3 (3.5P)		High permeability cores	1235	74.5	64.6	70.2	70.5	72.6
		Low permeability core	352	25.5	36.4	29.8	29.5	27.4

**Table 4 gels-08-00396-t004:** Experimental data on oil-increasing and precipitation effect of the chemical flooding model “profile control + plugging removal” of recovery factor.

	Parameter	Oil Saturation (%)	Recovery (%)				Recovery Factor Increase (%)		
Package Number			Water Flooding	The Chemical Flooding	Profile	Broken Down	Profile	Broken Down	The Overall
3-1	High permeability cores	71.51	36.10	62.56	-	63.26	-	0.70	27.16
	Low permeability core	70.75	8.34	15.24	-	15.92	-	0.68	7.58
	model	70.49	26.22	45.65	-	46.34	-	0.69	20.12
3-2	High permeability cores	70.79	34.82	61.65	67.80	68.54	6.15	0.74	33.72
	Low permeability core	70.25	8.50	16.91	23.16	25.76	6.25	2.60	17.26
	model	70.60	25.40	45.63	51.81	53.22	6.18	1.41	27.82
3-3	High permeability cores	71.77	35.28	62.03	70.88	72.58	8.58	1.70	37.30
	Low permeability core	70.85	8.24	17.20	27.42	31.63	10.22	4.21	23.39
	model	71.12	25.60	45.98	55.32	57.91	9.34	2.59	32.31

**Table 5 gels-08-00396-t005:** Plugging effect statistics of injection Wells.

Construction Time (Year)	Well No.	Broken Down Before			After the Broken Down		The Period of Validity (d)
		Daily Water Injection (m^3^/d)	The Water Injection Pressure (MPa)	Quantity of Injection Allocation (m^3^/d)	Daily Water Injection (m^3^/d)	The Water Injection Pressure (MPa)	
2018	E2-6	598	11.5	566	575	10.5	0
	E3-3	352	7.4	338	342	5.0	18
	A13	131	12.0	167	170	11.1	17
	D24	375	11.1	361	365	5.0	99
	D22	302	10.0	304	307	4.5	16
	E2-2	634	10.5	649	658	11.0	0
	E1-6	322	12.0	317	323	3.3	319
	W6-4	237	13.4	349	351	13.4	9
	W8-4	251	13.4	499	118	10.8	0
	W8-6	192	13.4	349	258	13.2	523
2019	W9-5	104	13.4	322	322	6.5	446
	W8-6	293	12.6	281	293	12.48	0
	W4-2	360	13.3	599	504	13.3	0
2020	W4-2	696	13.3	871	654	12.8	0

**Table 6 gels-08-00396-t006:** Basic parameters of the model.

Reservoir Parameters	Formation Pressure (MPa)	17.1
	Rock Compressibility (1/kPa)	1 × 10^−6^
Fluid parameters	Formation temperature (°C)	60
	Subsurface crude oil viscosity (mPa·s)	17.6
	Oil saturation pressure (MPa)	14
	Crude volume coefficient	1.1
	Formation water density (kg/m^3^)	1000
	Formation water viscosity (mPa·s)	0.47

**Table 7 gels-08-00396-t007:** Average polymer concentration in the vicinity of the wellbore.

The Serial Number	Construction Method of Plugging Removal	State of the Reservoir	Average Polymer Concentration (g·mol/m^3^)	
			Low Permeability Layer	High Permeability Layer
1	General broken down	Broken down before	50.7	35.6
		After the broken down	36.6	7.8
2	Profile control + plugging	Broken down before	50.7	35.6
		After the broken down	5.9	35.4

**Table 8 gels-08-00396-t008:** Relationship between injection rate and oil increase rate.

	Project	Package Number				
Parameter		Plan 3	Plan 4	Plan 5	Plan 6	Plan 7
Depolymerization system injection amount (m^3^)		60	80	100	120	140
Increased amount of oil (m^3^)		1084	1265	1512	1556	1650

**Table 9 gels-08-00396-t009:** Relationship between depolymerization agent concentration and oil increase.

	Project	Package Number					
Parameter		Plan 9	Plan 10	Plan 11	Plan 12	Plan 13	Plan 14
Concentration of depolymerization agent (mg/L)		500	1000	2000	4000	8000	12,000
Increased amount of oil (m^3^)		1319	1512	2160	2639	5330	5913

**Table 10 gels-08-00396-t010:** Relation between soaking time and oil increase.

	Project	Package Number				
Parameter		Plan 15	Plan 16	Plan 17	Plan 18	Plan 19
Soak time (h)		6	12	18	24	48
Increased amount of oil (m^3^)		4814	5330	5400	5427	5500

**Table 11 gels-08-00396-t011:** Details of pharmaceutical and construction prices.

The Serial Number	The Parameter Name	The Numerical
1	Forage-livestock system (%)	98.36
2	The VAT (%)	17
3	Profile control agent (Y/t)	4300
4	Plugging agent solution (Y/t)	35,300
5	Handling fee (ten thousand yuan)	40
6	Artificial cost (ten thousand yuan)	10
7	Equipment cost (ten thousand yuan)	25

**Table 12 gels-08-00396-t012:** Solvent water ion composition.

Cation			Anions				Total Mineralisation (mg/L)
Ca^2+^	Mg^2+^	Na^+^	CO_3_^2−^	HCO_3_^−^	Cl^−^	SO_4_^2−^	
50.75	19.05	1407	84	1741.42	1145.66	3.96	4451.84

**Table 13 gels-08-00396-t013:** Experimental programme content.

Scheme Number	Polymer Flooding	Plugging Removal Method	Injection Mode	Injection Pressure	Subsequent Water Flooding
1-1	“Constant speed” injection 0.5 PV	0.02 PV Plugging remover	Constant speed (0.6 mL/min)	-	Subsequent water flooding to 98% water cut
1-2	“Constant speed constant pressure” injection 0.5 PV		Constant pressure	2.5P	
2-1				2.5P	
2-2				3.0P	
2-3				3.5P	
3-1				2.5P	
3-2		0.06 PV Profile control of inorganic gel +0.02 PV Plugging removal		3.0P	
3-3		0.12 PV Profile control of inorganic gel +0.02 PV Plugging removal			

## Data Availability

The data presented in this study are available in the article.

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
