# Peer review of "Effect Evaluation and Action Mechanism Analysis of “Profile Control + Plugging Removal” after Chemical Flooding"

_gels, 2022, doi:10.3390/gels8070396_

Round 1

Reviewer 1 Report

The manuscript can be considered for publication as the topic quite interesting. However, some issues need to be addressed by the authors prior to acceptance for publication:

  1. The abstract is lengthy. Can make it more concise and directly address the gist of the study.
  2. Need to revise the keywords as some of the current keywords are not the common one.
  3. It is suggested to add relevant figure, photo or diagram to support the introduction and explanation about the field.
  4. Paper structure should be Introduction, Materials and Methods, Results and Discussion and then Conclusion. Please correct the structure of the paper.
  5. Problem statement is not clearly discussed and should be specifically addressed. The research gap is not well defined as well, thus does not provide good direction towards the objective of the study.
  6. The authors should clearly mention the gap based on the latest literature.
  7. What is the difference of this study with conformance control or water control by polymer gel?
  8. Why do the authors use this type of polymer? The authors should mention strong justification.
  9. Can you add more details about the polymer molecular weight so we can have a rough estimate of how large the molecule can be.
  10. Detail explanation of gel preparation should be added. What is the triggering factor for the gel to be formed? What pH was used to crosslink the gel?
  11. More in-depth discussion is required especially to answer the question of ‘why does it happen?’, to clarify and justify the results rather than just simply reporting the results.
  12. The quality of some figures and text on it are not in the best quality and not clearly visible. The resolution and size need improvement.
  13. The conclusion needs improvement. The conclusion can be made more concise and directly address the objective and problem statement.
  14. An aesthetic factor is the length of paragraphs which should not be too short or too long.
  15. More up-to-date references should be added, especially the study within 3 years back.
  16. Most of the references are from one country. Should include the study by authors from other countries or regions.
  17. Presentation in term of style, organization and format of the article need major improvement.
  18. There are some typos, grammatical errors, poor sentence structures and referencing errors that need to be proofread and corrected.

Author Response

Response to Reviewer 1 Comments

Point 1: The abstract is lengthy. Can make it more concise and directly address the gist of the study.

Response 1: Thank you for your suggestions. The abstract has been modified.

Abstract: The existing plugging removal operation in JZ9 - 3 oilfield has the disadvantages of small amount of plugging remover, fast injection speed and short construction time. Under the condition of injection well suction profile reversal, plugging remover is difficult to enter the low permeability part and play the role of deep plugging removal. In order to improve the plugging removal effect, this paper uses the physical simulation method to carry out the exper-imental study and mechanism analysis on the effect of water flooding, chemical flooding and plugging removal measures of multi-layer system combination model. The results show that the recovery of general plugging removal after chemical flooding increases by only 0.70 %, while the recovery of 'profile control + plugging removal' increases by '9.34 % + 2.59 %', and the amount of produced liquid decreases by more than 40 %. It can be seen that the combined opera-tion of profile control and plugging removal has dual effects of plugging and dredging and syn-ergistic effect, which not only expands the swept volume, but also reduces the inefficient and in-effective cycles. On this basis, the optimization design and effect prediction of the target well W4-2 plugging removal scheme are carried out by using numerical simulation method. Rec-ommended scheme: inorganic gel profile control agent volume 13243.6 m3, produced by the main agent (Na2O•nSiO2), spacer ( water ) and auxiliary agent (CaCl2) through multiple rounds of alternating injection into the reservoir. The plug removal agent (K2S2O8) injection volume is 100m3, the concentration is 0.8 %. The post-implementation ' Output / Input ' ratio is expected to be 3.7.

Point 2: Need to revise the keywords as some of the current keywords are not the common one.

Response 2: Thank you for your suggestions. The keyword section has been modified.

Keywords: JZ9-3 oilfield ; general plugging removal ; profile control + plugging removal ; EOR ; mechanism analysis

Point 3: It is suggested to add relevant figure, photo or diagram to support the introduction and explanation about the field.

Response 3: Thank you for your suggestions. The relevant figure, photo or diagram sections have been added in the end.

3.10 Mechanism analysis of "general plugging removal" and "profile control + plugging removal"

The ultimate goal of "general plugging removal" and "profile control + plugging removal" is to enhance oil recovery. The action mechanism of "general plugging removal" and "profile control + plugging removal" is shown in Figure 12 and Figure 13.

P0

P3

P2

P1

(1) After polymer flooding (2) plugging removal (3) plugging removal + subsequent water flooding

Fig. 12 General plugging removal mechanism

(1) After polymer flooding  (2) Profile control

(3) Profile control + plug removal  (4) Subsequent water drive

Fig. 13 action mechanism of "profile control + plug removal"

Point 4: Paper structure should be Introduction, Materials and Methods, Results and Discussion and then Conclusion. Please correct the structure of the paper.

Response 4: Thanks for the referee’s suggestion. The Paper structure have been revised.

Point 5: Problem statement is not clearly discussed and should be specifically addressed. The research gap is not well defined as well, thus does not provide good direction towards the objective of the study.

Response 5: Thanks for the referee’s suggestion. The problem statement have been revised.

Although chemical flooding in JZ9-3 oilfield has achieved a good effect of oil increase and precipitation, the water wells in the middle and later stages of chemical flooding and subsequent water flooding have experienced "Reverse rotation of liquid absorption profile", resulting in decreased liquid absorption and intensified inefficient and ineffective circula-tion [1]. "Liquid absorption profile reversal" means that the middle and low permeability layers (parts) of the reservoir are seriously damaged due to polymer retention. In order to remove the blockage and improve the liquid absorption profile, the oil field has succes-sively used acid, hydrogen peroxide, biological enzyme, solid chlorine dioxide, benzoyl peroxide, ammonium persulfate and potassium persulfate to remove the blockage of wa-ter wells. Good injection increase effect has been seen in the initial stage, but the mainte-nance time is short, and the oil increase effect at the oil well end is generally poor [2-10]. As the liquid suction pressure difference, i.e. the liquid suction capacity, at the high per-meability part of the reservoir is much greater than that at the middle and low permeabil-ity part, and in addition, the plugging removal dose used in the construction operation is less, the liquid injection speed is faster, and the construction time is shorter, the plugging removal agent suction at the middle and low permeability part of the reservoir is very small, and the plugging removal effect is poor [11-16]. In the oilfield production practice, increasing the liquid suction pressure difference is the only way to increase the suction amount of plugging removal agent in the middle and low permeability parts and improve the plugging removal effect. Increasing the injection pressure is an important measure to increase the liquid suction pressure difference [17-22]. Increasing the injection pressure can be achieved by increasing the injection speed and increasing the seepage resistance at the high permeability part, i.e. the start-up pressure of liquid absorption. The former will have poor effect due to the influence of process conditions and the "flow around" of fluid in the reservoir, while the latter can increase the injection pressure without increasing the injection speed [23]. Therefore, "profile control + plugging removal" is one of the effective technical ways to further enhance oil recovery after chemical flooding. In order to realize the effective plugging removal of the middle and low permeability layer in the polymer plugging well, according to the actual situation existing in the production process of J oil-field, the parallel core physical simulation experiment was used to explore the influence of the flooding agent injection mode, the injection pressure of the depolymerization system and the joint operation of "profile control + plugging removal" on the plugging removal effect. The scheme design and effect prediction are carried out for difficult wells. The slug size of profile control agent is optimized and the economic benefit is evaluated by using numerical simulation technology. Good results are obtained in the experimental process, which improves the technical support for the application of plugging removal process measures in oil fields.

Point 6: The authors should clearly mention the gap based on the latest literature.

Response 6: Thank you for your suggestions. The latest literature have been revised.

[1]Chen, Huaxing , Wang, Yufei , Pang, Ming , Fang, Tao , Zhao, Shunchao , Wang, Zhiyuan , and Yugang Zhou. Research on Plugging Mechanism and Optimisation of Plug Removal Measure of Polymer Flooding Response Well in Bohai Oilfield. Paper presented at the International Petroleum Technology Conference, Virtual, March 2021. doi: https://doi.org/10.2523/IPTC-21271-MS

[2]Liu, He , Yang, Gao , and Liu Fangchao. An Overview of Polymer Broken Down for Increasing Injectability in Polymer Flooding. Paper presented at the SPE Asia Pacific Oil and Gas Conference and Exhibition, Jakarta, Indonesia, October 2013. doi: https://doi.org/10.2118/165825-MS

[3]Hoffmann, Falk,Machatschek, Rainhard,Lendlein, Andreas. Analytical model and Monte Carlo simulations of polymer degradation with improved chain cut statistics[J]. Journal of Materials Research,2022(prepublish).

[4]Sun Guoqiang,Zhu Xiaobin,Zhang Qiyi,Yan Chuanqi,Ning Weidong,Wang Tao. Oxidation and polymer degradation characteristics of high viscosity modified asphalts under various aging environments[J]. Science of the Total Environment,2022,813.

[5]Yang Chen,Lu Yu,Cao Liping,Liu Zhiyang,Zhang Tiantao,Yu Hao,Zhang Bingtao,Dong Zejiao. Polymer degradation mechanism and chemical composition relationship of hot-poured asphaltic crack repair material during thermal aging exploiting fluorescence microscopy and gel permeation chromatography[J]. Construction and Building Materials,2021,302.

[6]Elbing Brian R.. Impact of Polymer Degradation On Past Studies of the Mean Velocity Profile in Turbulent Boundary Layers[J]. J. Fluids Eng,2021.

[7]Vohlídal Jiří. Polymer degradation: a short review[J]. Chemistry Teacher International,2020,3(2).

[8]Celina Mathew C.,Linde Erik,Martinez Estevan. Carbonyl Identification and Quantification Uncertainties for Oxidative Polymer Degradation[J]. Polymer Degradation and Stability,2021(prepublish).

[9]P. Druetta,F. Picchioni. Influence of the polymer degradation on enhanced oil recovery processes[J]. Applied Mathematical Modelling,2019,69.

[10]Aishwarya Kulkarni,Harshini Dasari. Current Status of Methods Used In Degradation of Polymers: A Review[J]. MATEC Web of Conferences,2018,144.

[11]Qingming Gan,Yi Qin,Wei Huang,Mei Lu,Ruiquan Liao. The Research of Ja Composite Acidizing Plugging Removal System: Formulation Optimization and Performance Evaluation[J]. Chemical Engineering Transactions (CET Journal),2017,62.

[12]Chao Wang,Changjun Long,Jianli Zhang,Jie Xiao,Zhengdong Xu,Feng Yan. Research on Targeted Acidizing Technology in Long Horizontal Section of Carbonate Reservoir[J]. E3S Web of Conferences,2021,233.

[13]Czupski Marek,Kasza Piotr,Leśniak Łukasz. Development of Selective Acidizing Technology for an Oil Field in the Zechstein Main Dolomite[J]. Energies,2020,13(22).

[11]Gang Sun, Changqing Li, Xinmin Zhang, et al. Effect of sodium dodecylbenzene sulfonate on rheological properties of hydrophobically associating polymer solutionsp[J]. Petroleum Geology & Oilfield Development in Daqing, 2012, 31 (5): 132-136

[12] Jianzhong Zhang, Jirui Hou, Hongjiao Tang, et al Injection and migration properties of CMC and acrylic polymer[J]. Petroleum Geology & Oilfield Development in Daqing, 2012,31 (6): 151-155

[13]Yanping Chu. Application of polymer surfactant in strong alkali composite flooding system[J]. Petroleum Geology & Oilfield Development in Daqing,2013,32(2):102-105.

[14]Jing Tang, Shangqi Shao, Wengang Ding, Ruoxuan Tang. Research Progress and Prospect of Acidizing Process and Acid Fracturing Technology[J]. Journal of Petroleum and Mining Engineering,2019,2(1).

[15]Ning Mengmeng,Che Hang,Kong Weizhong,Wang Peng,Liu Bingxiao,Xu Zhengdong,Wang Xiaochao,Long Changjun,Zhang Bin,Wu Youmei. Research and application of multi-hydrogen acidizing technology of low-permeability reservoirs for increasing water injection[J]. IOP Conference Series: Earth and Environmental Science,2017,100.

[16]Shao Hui He. Study on Damage and Solution of Acidizing Technology in Sandstone Reservoir[J]. Advanced Materials Research,2013,2200(631-632).

[17] Xie Kun, Cao Bao, Lu Xiangguo, et al. Matching between the diameter of the aggregates of hydrophobically associating polymers and reservoir pore-throat size during polymer flooding in an offshore oilfield[J]. Journal of petroleum science &engineering, 2019, 177: 558-569,

[18]Jing Shu Yuan,Guang Sheng Cao,Ping Chen,Yang Gao. A Kind of Novel Chemical Plugging Removal Technology Study for Polymer Flooding Wells[J]. Advanced Materials Research,2014,2954(884-885).

[19]Wei Lili,Xie Juan,Yu Jiliang,Cai Shaosong,Zhang Wei,Yan Haijun,Ma Zhi,Su Na,Gao MengXiang,Liu Shasha,Qin Zhonghai. Evaluation and Application of New Anti-expansion Unblocking Agent for Water-sensitive Oil Deposit[J]. IOP Conference Series: Earth and Environmental Science,2019,371.

[20]V.R.S. De Silva,P.G. Ranjith,M.S.A. Perera,B. Wu,T.D. Rathnaweera. A modified, hydrophobic soundless cracking demolition agent for non-explosive demolition and fracturing applications[J]. Process Safety and Environmental Protection,2018,119.

[21]Yiyong Pan,Changlong Liu,Liqiang Zhao,Feng Li,Yigang Liu,Pingli Liu. Study and application of a new blockage remover in polymer injection wells[J]. Petroleum,2018,4(3).

[22] Xie Kun, Lu Xiangguo, Pan He, et al. Analysis of dynamic imbibition effect of surfactant in micro cracks in reservoir with high temperature and low permeability[J]. SPE Production & Operations, 2018, 33(3): 596-606.

[23]Cao Jing Ping,Zhao Fu Lei,Zhang Kun. Reasons for Formation Damage and Preparation for Disentanglement Agent in Low Permeability Oilfield[J]. Applied Mechanics and Materials,2013,457-458(457-458).

Point 7: What is the difference of this study with conformance control or water control by polymer gel?

Response 7: Thank you for your suggestions.

It can be seen from Fig. 12 and FIG. 13 that in the process of polymer flooding, firstly, the polymer enters the high-permeability layer with low seepage resistance and stays in the pore throat (chemical adsorption and mechanical capture), resulting in the increase of seepage resistance in the high-permeability layer, the decrease of suction pressure difference and the increase of injection pressure. When the injection pressure is greater than the start-up pressure of liquid absorption in the medium and low permeability layer, the medium and low permeability layer begins to absorb liquid, that is, the fluid flow turns in the reservoir. At the same time, the seepage resistance of the middle and low permeability layer increases, but the increase rate of the seepage resistance is greater than that of the high permeability layer. At this time, the suction profile is reversed, and the high permeability layer starts to suck again. It is worth noting that the polymer will cause damage to the medium and low permeability layer after entering the medium and low permeability layer. Therefore, it is necessary to carry out plugging removal for the medium and low permeability layer. The commonly used plugging removal method in oil field is "general plugging removal". That is, in the late stage of polymer flooding, plugging remover is directly injected, and the plugging remover enters the high permeability layer along the part with relatively small seepage resistance. During the well plugging process, it reacts with the polymer retained in the high permeability layer, further reducing the seepage resistance of the high permeability layer and increasing the subsequent inefficient and ineffective circulation of water in the high permeability layer. "Profile control + plugging removal" means that after polymer flooding, an inorganic gel with a small slug size is alternately injected as a profile control agent to form a small range of plugging near the well zone of the high permeability layer, which increases the seepage resistance at the inlet of the high permeability layer and near the well zone, the injection pressure, the liquid absorption pressure difference in the middle and low permeability layer, and the relative liquid absorption volume. Then inject the plugging removal agent to make it oxidize and degrade with the polymer retained in the plugging part near the well in the middle and low permeability layer. The long chain of the polymer is oxidized and degraded into a short chain, which increases the liquid flow capacity of the middle and low permeability layer, further reduces the seepage resistance of subsequent water in the middle and low permeability layer, and increases the liquid suction pressure difference and relative liquid suction capacity of the middle and low permeability layer.

Point 8: Why do the authors use this type of polymer? The authors should mention strong justification.

Response 8: Thank you for pointing out this problem. The reasons for using this polymer have been added.

In order to simulate the plugging problem of salt-resistant polymer in offshore oilfield, the salt-resistant polymer used in JZ9-3 oilfield is used in the experiment.

Point 9: Can you add more details about the polymer molecular weight so we can have a rough estimate of how large the molecule can be.

Response 9: Thank you for pointing out this problem. The polymer molecular weight has been added.The polymer is polymer SNF3640 with effective content of 90 % and molecular weight of 19 million, which is provided by CNOOC Tianjin Branch.

Point 10: Detail explanation of gel preparation should be added. What is the triggering factor for the gel to be formed? What pH was used to crosslink the gel?

Response 10: Thank you for pointing out this problem. The explanation of gel preparation has been added.

3.3 Evaluation of basic properties of inorganic gel

4% sodium silicate and 4% calcium chloride are prepared with simulated injection water from J oilfield. 50ml of each is added to a 100ml measuring cylinder and placed in a 65 ℃ incubator. The relationship between the standing time and the volume of inorganic gel is shown in Figure 4.

Na2SiO3 +CaCl2 = CaSiO3↓ +2NaCl

Fig. 4 Relationship between standing time and inorganic gel system

It can be seen from Figure 4 that the main agent reacts with the auxiliary agent to form inorganic gel. With the increase of the standing time, the volume of inorganic gel gradually decreases. The analysis shows that the inorganic gel reaction "coagulates at one touch". With the increase of the standing time, the inorganic gel settles, which can fo

rm an inorganic coating in the porous medium, reduce the overflow section, and promote the subsequent plugging removal system to enter the medium and low permeability layer. At the same time, sodium silicate is alkaline, which can convert the amino group of the polymer into carboxyl group, and further increase the flow capacity of the polymer.

Point 11: More in-depth discussion is required especially to answer the question of ‘why does it happen?’, to clarify and justify the results rather than just simply reporting the results.

Response 11: Thank you for pointing out this problem. The discussion has been added.

Point 12: The quality of some figures and text on it are not in the best quality and not clearly visible. The resolution and size need improvement.

Response 12: Thank you for pointing out this problem. The quality of some figures and text have been revised.

Point 13: The conclusion needs improvement. The conclusion can be made more concise and directly address the objective and problem statement.

Response 13: Thank you for pointing out this problem. The conclusion has been modified.

Point 14: An aesthetic factor is the length of paragraphs which should not be too short or too long.

Response 14: Thank you for your suggestion. Article paragraph has been changed.

Point 15: More up-to-date references should be added, especially the study within 3 years back.

Response 15: Thank you for your suggestions. The latest references have been revised.

[1]Chen, Huaxing , Wang, Yufei , Pang, Ming , Fang, Tao , Zhao, Shunchao , Wang, Zhiyuan , and Yugang Zhou. Research on Plugging Mechanism and Optimisation of Plug Removal Measure of Polymer Flooding Response Well in Bohai Oilfield. Paper presented at the International Petroleum Technology Conference, Virtual, March 2021. doi: https://doi.org/10.2523/IPTC-21271-MS

[2]Liu, He , Yang, Gao , and Liu Fangchao. An Overview of Polymer Broken Down for Increasing Injectability in Polymer Flooding. Paper presented at the SPE Asia Pacific Oil and Gas Conference and Exhibition, Jakarta, Indonesia, October 2013. doi: https://doi.org/10.2118/165825-MS

[3]Hoffmann, Falk,Machatschek, Rainhard,Lendlein, Andreas. Analytical model and Monte Carlo simulations of polymer degradation with improved chain cut statistics[J]. Journal of Materials Research,2022(prepublish).

[4]Sun Guoqiang,Zhu Xiaobin,Zhang Qiyi,Yan Chuanqi,Ning Weidong,Wang Tao. Oxidation and polymer degradation characteristics of high viscosity modified asphalts under various aging environments[J]. Science of the Total Environment,2022,813.

[5]Yang Chen,Lu Yu,Cao Liping,Liu Zhiyang,Zhang Tiantao,Yu Hao,Zhang Bingtao,Dong Zejiao. Polymer degradation mechanism and chemical composition relationship of hot-poured asphaltic crack repair material during thermal aging exploiting fluorescence microscopy and gel permeation chromatography[J]. Construction and Building Materials,2021,302.

[6]Elbing Brian R.. Impact of Polymer Degradation On Past Studies of the Mean Velocity Profile in Turbulent Boundary Layers[J]. J. Fluids Eng,2021.

[7]Vohlídal Jiří. Polymer degradation: a short review[J]. Chemistry Teacher International,2020,3(2).

[8]Celina Mathew C.,Linde Erik,Martinez Estevan. Carbonyl Identification and Quantification Uncertainties for Oxidative Polymer Degradation[J]. Polymer Degradation and Stability,2021(prepublish).

[9]P. Druetta,F. Picchioni. Influence of the polymer degradation on enhanced oil recovery processes[J]. Applied Mathematical Modelling,2019,69.

[10]Aishwarya Kulkarni,Harshini Dasari. Current Status of Methods Used In Degradation of Polymers: A Review[J]. MATEC Web of Conferences,2018,144.

[11]Qingming Gan,Yi Qin,Wei Huang,Mei Lu,Ruiquan Liao. The Research of Ja Composite Acidizing Plugging Removal System: Formulation Optimization and Performance Evaluation[J]. Chemical Engineering Transactions (CET Journal),2017,62.

[12]Chao Wang,Changjun Long,Jianli Zhang,Jie Xiao,Zhengdong Xu,Feng Yan. Research on Targeted Acidizing Technology in Long Horizontal Section of Carbonate Reservoir[J]. E3S Web of Conferences,2021,233.

[13]Czupski Marek,Kasza Piotr,Leśniak Łukasz. Development of Selective Acidizing Technology for an Oil Field in the Zechstein Main Dolomite[J]. Energies,2020,13(22).

[11]Gang Sun, Changqing Li, Xinmin Zhang, et al. Effect of sodium dodecylbenzene sulfonate on rheological properties of hydrophobically associating polymer solutionsp[J]. Petroleum Geology & Oilfield Development in Daqing, 2012, 31 (5): 132-136

[12] Jianzhong Zhang, Jirui Hou, Hongjiao Tang, et al Injection and migration properties of CMC and acrylic polymer[J]. Petroleum Geology & Oilfield Development in Daqing, 2012,31 (6): 151-155

[13]Yanping Chu. Application of polymer surfactant in strong alkali composite flooding system[J]. Petroleum Geology & Oilfield Development in Daqing,2013,32(2):102-105.

[14]Jing Tang, Shangqi Shao, Wengang Ding, Ruoxuan Tang. Research Progress and Prospect of Acidizing Process and Acid Fracturing Technology[J]. Journal of Petroleum and Mining Engineering,2019,2(1).

[15]Ning Mengmeng,Che Hang,Kong Weizhong,Wang Peng,Liu Bingxiao,Xu Zhengdong,Wang Xiaochao,Long Changjun,Zhang Bin,Wu Youmei. Research and application of multi-hydrogen acidizing technology of low-permeability reservoirs for increasing water injection[J]. IOP Conference Series: Earth and Environmental Science,2017,100.

[16]Shao Hui He. Study on Damage and Solution of Acidizing Technology in Sandstone Reservoir[J]. Advanced Materials Research,2013,2200(631-632).

[17] Xie Kun, Cao Bao, Lu Xiangguo, et al. Matching between the diameter of the aggregates of hydrophobically associating polymers and reservoir pore-throat size during polymer flooding in an offshore oilfield[J]. Journal of petroleum science &engineering, 2019, 177: 558-569,

[18]Jing Shu Yuan,Guang Sheng Cao,Ping Chen,Yang Gao. A Kind of Novel Chemical Plugging Removal Technology Study for Polymer Flooding Wells[J]. Advanced Materials Research,2014,2954(884-885).

[19]Wei Lili,Xie Juan,Yu Jiliang,Cai Shaosong,Zhang Wei,Yan Haijun,Ma Zhi,Su Na,Gao MengXiang,Liu Shasha,Qin Zhonghai. Evaluation and Application of New Anti-expansion Unblocking Agent for Water-sensitive Oil Deposit[J]. IOP Conference Series: Earth and Environmental Science,2019,371.

[20]V.R.S. De Silva,P.G. Ranjith,M.S.A. Perera,B. Wu,T.D. Rathnaweera. A modified, hydrophobic soundless cracking demolition agent for non-explosive demolition and fracturing applications[J]. Process Safety and Environmental Protection,2018,119.

[21]Yiyong Pan,Changlong Liu,Liqiang Zhao,Feng Li,Yigang Liu,Pingli Liu. Study and application of a new blockage remover in polymer injection wells[J]. Petroleum,2018,4(3).

[22] Xie Kun, Lu Xiangguo, Pan He, et al. Analysis of dynamic imbibition effect of surfactant in micro cracks in reservoir with high temperature and low permeability[J]. SPE Production & Operations, 2018, 33(3): 596-606.

[23]Cao Jing Ping,Zhao Fu Lei,Zhang Kun. Reasons for Formation Damage and Preparation for Disentanglement Agent in Low Permeability Oilfield[J]. Applied Mechanics and Materials,2013,457-458(457-458).

Point 16: Most of the references are from one country. Should include the study by authors from other countries or regions.

Response 16: Thank you for your suggestion. The reference has been revised.

Point 17: Presentation in term of style, organization and format of the article need major improvement.

Response 17: Thank you for your suggestion. We have made an overall adjustment to the full text.

Point 18: There are some typos, grammatical errors, poor sentence structures and referencing errors that need to be proofread and corrected.

Response 18: Thank you for your suggestion. We have corrected the grammatical errors.

Reviewer 2 Report

  1. The title needs to be revised and checked.
  2. Although the paper is well written concerning the application section, the authors should discuss in detail some of the physical and chemical analyses related to the used polymer including (FT-IR, H-NMR, TGA, Molecular weight,) tests in addition to rheological properties of the used polymer solution at simulated flooding conditions.

Author Response

Response to Reviewer 1 Comments

Point 1: The title needs to be revised and checked.

Response 1: Thank you for your suggestions. The title has been revised

Point 2: Although the paper is well written concerning the application section, the authors should discuss in detail some of the physical and chemical analyses related to the used polymer including (FT-IR, H-NMR, TGA, Molecular weight,) tests in addition to rheological properties of the used polymer solution at simulated flooding conditions.

Response 2: Thank you for your suggestions.The FTIR and rheological properties have been added.

Reviewer 3 Report

The authors present a comprehensive field study on the sweep efficiency of non-uniform reservoirs regarding the permeability and the attempt to plug high permeable areas in order to sweep lower permeable zones by using polymer flooding. The real situation is simulated in a lab setup using two cores of different permeability in parallel. The procedures are described in detail. The work is merely descriptive and apparently, the authors do not manage to substantially improve the sweep efficiency.

The figures are of bad quality, not readable like Figs. 9 to 11 (Fig. 11 does not even have a capture) and partly not understandable, like Figure 3 that contains chinese descriptions. The difference between the upper and lower diagrams is not clear. Descriptions and definitions are missing, e.g. what is meant by shunt rate and what are the numbers 0.6 ml/min etc.

Fig. 12 is not described at all. How is the rel perm obtained and what is the interpretation in the context of the current study?

The conclusion of the entire work is limited to describing how inefficient the production in JZ9-3 was and that the method applied was not successful. The reader would be interested in a hypothesis of why this is so and what could be done otherwise. There is no consistent approach to the possible physico-chemical mechanisms presented, e.g. how could a viscoelastic surfactant help to block larger pores and leave smaller pores open? The work is concluded with the trivial statement that “a good injection and plugging performance … is the technical guarantee for the successful … operation …”  

Author Response

Response to Reviewer 3 Comments

Point 1: The authors present a comprehensive field study on the sweep efficiency of non-uniform reservoirs regarding the permeability and the attempt to plug high permeable areas in order to sweep lower permeable zones by using polymer flooding. The real situation is simulated in a lab setup using two cores of different permeability in parallel. The procedures are described in detail. The work is merely descriptive and apparently, the authors do not manage to substantially improve the sweep efficiency.

Response 1: Thank you for pointing out this problem. In view of the insufficient analysis of job description results, further discussion and analysis are added at the end of this paper. The answer to why there is no significant EOR is as follows. Firstly, physical simulation experiments are designed to simulate the real reservoir development. The simple plugging removal and profile control and plugging removal are compared. It can be seen from the results that under the same permeability conditions, the development mode of "profile control + plugging removal" is better than simple plugging removal and profile control, and the EOR effect is better. Secondly, the effect of EOR in the whole process is limited, mainly because the size of traditional Chinese medicine injection slug in the field injection process is small, so the size of injection slug in the experiment is designed to be small. Finally, the purpose of this paper is to remove the blockage of low and medium permeability parts. If you want to continue to improve oil recovery in the later stage, you can take relevant measures of profile control / displacement control.

Point 2: The figures are of bad quality, not readable like Figs. 9 to 11 (Fig. 11 does not even have a capture) and partly not understandable, like Figure 3 that contains chinese descriptions. The difference between the upper and lower diagrams is not clear. Descriptions and definitions are missing, e.g. what is meant by shunt rate and what are the numbers 0.6 ml/min etc.

Response 2: Thank you for your suggestions. Relevant charts have been modified. Relevant definitions have been added to the manuscript.

The diversion rate is divided into high permeability layer diversion rate and low permeability layer diversion rate. The diversion rate of high permeability layer is the ratio of produced fluid to total produced fluid. The diversion rate of the low permeability layer is the ratio of the produced liquid volume of the low permeability layer to the total pro-duced liquid volume.

Point 3: Fig. 12 is not described at all. How is the rel perm obtained and what is the interpretation in the context of the current study?

Response 3:  Thank you for your suggestions. The relative permeability curve is obtained from the data provided by the oil field, which determines the law of oil-water flow in the process of numerical simulation.

Point 4: The conclusion of the entire work is limited to describing how inefficient the production in JZ9-3 was and that the method applied was not successful. The reader would be interested in a hypothesis of why this is so and what could be done otherwise. There is no consistent approach to the possible physico-chemical mechanisms presented, e.g. how could a viscoelastic surfactant help to block larger pores and leave smaller pores open? The work is concluded with the trivial statement that “a good injection and plugging performance … is the technical guarantee for the successful … operation …”

Response 4: Thank you for your question. Relevant papers have been added in this paper. The conclusion has also been modified.

Round 2

Reviewer 1 Report

The manuscript can be accepted for publication. However, minor formatting correction need to be done prior final version. The current arrangement and form of the article are quite difficult for first time reader to comprehend.

Reviewer 2 Report

the manuscript can be published in its current form

Reviewer 3 Report

The manuscript has been improved and can be accepted for publication in this journal.